# The long-term impact of the Covid-19 pandemic on financial insecurity in vulnerable families: Findings from the Born in Bradford Covid-19 longitudinal study

Sian Reece[1]*, Josie Dickerson[2], Brian Kelly[2], Rosemary R. C. McEachan[2], Kate E. Pickett[3]

1 Hull York Medical School, York, North Yorkshire, United Kingdom, 2 Bradford Institute for Health Research, Bradford Royal Infirmary, Bradford, West Yorkshire, United Kingdom, 3 Department of Health Services, University of York, Heslington Road, York, North Yorkshire, United Kingdom

* sianreece@doctors.org.uk

**Data Availability Statement:** This data is available through a system of managed open access. Researchers who would like access to this data, or

## Abstract

There is growing recognition that the public health measures employed to control the spread of the COVID-19 pandemic had unintended consequences on socioeconomic security and health inequalities, having the greatest impact on the most vulnerable groups. This longitudinal study aims to explore the medium to long-term impacts of the COVID-19 pandemic and subsequent public health measures on financial security for families living in the deprived and ethnically diverse city of Bradford. We collected data at four time points before and during the pandemic from mothers who participated in one of two prospective birth cohort studies in Bradford. The findings demonstrate that the risk of experiencing financial insecurity rose sharply during the pandemic and has not returned to pre-COVID-19 baseline levels. Several individual characteristics were found to be possible predictors of financial insecurity, including homeowner status, free school meal eligibility and not working. Protective factors against financial insecurity include: living in more affluent areas; greater levels of educational attainment; and families with two or more adults in the household. Notably, families of Pakistani Heritage were found to have the greatest risk of experiencing financial insecurity throughout the pandemic. Furthermore, this study demonstrated that there were strong associations between financial insecurity and maternal health and wellbeing outcomes, with mothers experiencing financial insecurity being more likely to report unsatisfactory general health and clinically important symptoms of depression and anxiety. The findings of this study highlight that the impact of financial insecurity experienced by mothers and their families throughout the pandemic was severe, wide ranging and affected the most vulnerable. In the wake of the pandemic, the emerging cost of living and energy crisis emphasises the urgent need for policy makers to act to support vulnerable families to prevent further widening of existing health and social inequalities.

any other Born in Bradford data, are encouraged to submit an expression of interest to borninbradford@bthft.nhs.uk, which will be reviewed by the BiB Executive who meet to review proposals on a monthly basis and will endeavour to respond to your request as soon as possible. If your request is approved we will ask you to sign a Data Sharing Contract and a Data Sharing Agreement. For further information please see: How to access data - Born In Bradford (https://borninbradford.nhs.uk/research/how-to-access-data/).

**Funding:** This study was funded by The Health Foundation (https://www.health.org.uk) COVID-19 Award (2301201), with further contributions from a Wellcome Trust (https://wellcome.org) infrastructure grant (WT101597MA); a joint grant from the UK Medical Research Council (MRC) (https://www.ukri.org/councils/mrc/) and UK Economic and Social Science Research Council (ESRC) (https://www.ukri.org/councils/esrc/) (MR/N024391/1); the National Institute for Health and Care Research (NIHR) (https://www.nihr.ac.uk) under its Applied Research Collaboration Yorkshire and Humber (NIHR200166); ActEarly UK Prevention Research Partnership Consortium (https://ukprp.org/what-we-fund/actearly/) (MR/S037527/1); Better Start Bradford through The National Lottery Community Fund (https://www.tnlcommunityfund.org.uk); and the British Heart Foundation (https://www.bhf.org.uk/) (CS/16/4/32482). SR is in receipt of an NIHR doctoral research fellowship (https://www.nihr.ac.uk) (NIHR300677). None of the sponsors or funders played a role in the study design, data collection and analysis, decision to publish, or preparation of the manuscript.

**Competing interests:** The authors have declared that no competing interests exist.

## Introduction

In response to the COVID-19 pandemic, the UK government, like many others internationally, implemented stringent lockdowns to slow the spread of the virus throughout the population and to limit the number of severe COVID-19 cases and consequent pressures on the National Health Service. During the first UK lockdown, implemented from March 23rd, 2020, this included the closure of all schools, non-essential shops and businesses, reduced health and social care provision and imposed restrictions on daily activities [1, 2]. Where possible people were advised to work from home. Key worker status was given to public and private-sector employees whose work was deemed critical to the COVID-19 response status [3]. These employees were permitted to travel to work, where necessary. The national furlough scheme was established to financially support employees placed on temporary leave, for some or all of their contracted hours, to ensure they received at least 80% of their usual wages whilst furloughed [4, 5].

## Literature review

There is growing recognition that the public health measures employed to control the spread of the COVID-19 pandemic had unintended consequences on socioeconomic security and have increased inequalities, with families from deprived and ethnically diverse backgrounds most likely to be adversely affected [6, 7]. However, many longitudinal studies that focussed on socioeconomic security conducted within the UK focused on participants of White European ancestry from relatively affluent populations and did not have pre-COVID-19 baseline data [8, 9].

The relationship between financial insecurity and poor health and wellbeing is well established. Early childhood deprivation is associated with significant negative physical, mental health and social outcomes that not only limit a child's development in the short-term but have long lasting effects into adulthood [10]. In adulthood, links between financial difficulties, social deprivation and mental health are also well established [11]. Financial insecurity can precipitate and perpetuate mental health problems [11, 12] and has been found to be a predictor of chronic physical illness [13–15]. Furthermore, individuals suffering with poor mental health associated with financial insecurity, worsened in recent years by austerity, are more likely to face challenges in accessing the advice and support needed to address these welfare issues [12, 15]. The World Health Organization estimates that income security accounts for almost two thirds of health inequities between socioeconomic groups within countries of the European region [16]. Furthermore, poor health can also lead to socioeconomic welfare problems, perpetuating the cycle of deteriorating health and socioeconomic welfare, and perpetuating inequalities.

## The Born in Bradford research programme

Born in Bradford (BiB) research programme is an internationally recognised, applied health research programme comprising health and wellbeing information on more than 30,000 Bradfordians enrolled in a family of three large, multi-ethnic prospective birth cohort studies: BiB Family; Born in Bradford's Better Start BiBBS; and BiB4All [17].

The aims of the research programme are fourfold: to describe health and ill-health in the largely bi-ethnic population with high economic deprivation; to identify modifiable causal relationships that contribute to ill-health, and design and evaluate interventions to promote wellbeing; to provide an integrated model of epidemiological and evaluative research based on practice in the National Health Service and related health systems; and to build and reinforce

research capacity in Bradford [18, 19]. The protocol for this study is described elsewhere [18, 19].

The BiB research programme provides in-depth longitudinal information on the demographics and socioeconomic and health status of mothers before the pandemic and at three time points during the pandemic to describe the trajectories and identify the long-term consequences of the pandemic on vulnerable populations. This offers a unique opportunity to assess the socioeconomic impact of the pandemic longitudinally on families in a highly ethnically diverse population, the majority of whom live in the most deprived centiles in the UK and in whom mental ill health is often reported to be more prevalent [20–22]. Data published from the first BiB COVID-19 survey during the first UK lockdown [23] found that more than one-third of families reported financial insecurity. Financial insecurity at that time was predicted by previous financial security, employment status and ethnicity. There were also strong associations found between financial insecurity and poor family relationships, mental health and negative health behaviours.

This study aims to explore the medium to long-term impacts of the COVID-19 and subsequent public health measures on financial security for families in Bradford and answers the following research questions:

a. What were the impacts of the COVID-19 pandemic and subsequent public health measures on levels of financial security for mothers in Bradford?

b. What individual factors were associated with changes to financial security during the pandemic?

c. What impact did changes to financial security have on the health, wellbeing and socioeconomic security of mothers during the pandemic?

## Methods

### Study design

A longitudinal study collected survey data at four time points before and during the COVID-19 pandemic from mothers who participated in one of two prospective cohort studies in Bradford: Born in Bradford's Growing Up (BiBGU) cohort study, with mothers of children aged 9–13 years [24, 25]; and Born in Bradford's Better Start (BiBBS) cohort study, with mothers of children aged 0–5 years old [26].

### Ethics

This study involves human participants and was approved by the HRA and Bradford/Leeds Research Ethics Committee (substantial amendments to BiBGU 16/YH/0320 and BiBBS 15/YH/0455).

Participants gave informed consent to participate in the study before taking part. Participants had previously consented for their research data, and routinely collected health and education data, to be used for research. For the COVID-19 survey, verbal consent was taken for questionnaires completed over the phone and implied consent was assumed for all questionnaires completed via post or online.

### Data collection

All participants from the BiBBS and BiBGU cohort studies were contacted to ask if they wished to participate in this study. Participants were recruited and data were collected using a

combination of methods, including emails, text and telephone, with a follow-up postal survey in order to facilitate a rapid response. Participants were recruited in their main language wherever possible. Full details of the data collection of the survey can be found elsewhere [23, 27]. The phase one survey was administered between April and June 2020, the phase two survey was administered between October and December 2020 and the phase three survey was administered between May and July 2021.

Pre-COVID-19 baseline levels of self-reported financial security and mental health outcomes for BiBGU participants were derived from two sources: levels of financial insecurity were collected during pregnancy between 2007–2011 [18]; and recent follow-up data on mental health were collected between 24th June 2017 and 12th March 2020 [24]. Pre-COVID-19 baseline data for BiBBS participants were taken from data collected during pregnancy between 6th January 2016 and 8th February 2020 [26]. The median time since most recent pre-COVID-19 data collection was 15 months (range 1 to 35 months) for BiBGU and 29 months (range 2 to 52 months) for BiBBS participants. Full details of the protocol and data collected for the BiBBS experimental birth cohort and the BiBGU cohort study are described in full elsewhere [24, 26].

## Patient and public involvement

Born in Bradford is a 'people powered' research study. The local community were consulted to identify key research priorities throughout the pandemic as a part of the BiB COVID-19 research programme. This included consultation with key community groups, seldom-heard communities and local policy and decision makers to ensure that the focus of the research was relevant to local needs. The COVID-19 survey and recruitment approach were tested through established community research advisory groups. The findings of the study were also shared with these groups to enhance interpretation and ensure useful dissemination back to the community. Full details can be found in the protocol paper [27].

## Outcome measures

Survey questions were selected from validated questionnaires, from previous Born in Bradford questionnaires or were devised specifically for this survey. The key domains were: household circumstances [28]; family relationships and social support [29–31]; financial security [32, 33]; and physical and mental health [34–37].

Ethnicity was coded using Census 2011 categories. The majority of mothers identified as 'White British' or of 'Pakistani Heritage'. There were small numbers of mothers from a number of other ethnic groups who did not identify as 'White British' and 'Pakistani Heritage'. The sample sizes for each of these other ethnic groups were too small to permit meaningful data analysis, therefore these mothers were grouped and categorised within the 'Other' ethnicity category. Residential address (as at 31st March 2019) was linked to the 2019 Index of Multiple Deprivation [38].

To establish financial insecurity, the surveys employed the question: 'How well would you say you are managing financially right now?'. Answer options included: living comfortably; doing alright; just about getting by; finding it quite difficult; and finding it very difficult. The latter two options were grouped and categorised as indicating financial insecurity.

For mental health, the Patient Health Questionnaire-8 (PHQ-8) and General Anxiety Disorder-7 (GAD-7) instruments were used. Standard categorisations were employed for the scores (0 to 4 no depression, 5 to 9 mild depression, 10–14 moderate depression, 15–24 severe depression; 0 to 4 no anxiety, 5 to 9 mild anxiety, 10 to 14 moderate anxiety, 15+ severe

anxiety) [36, 37]. Clinically important symptoms of depression and anxiety were defined as those with moderate to severe anxiety and depression respectively.

To explore the effect of the pandemic related changes to financial security on the socioeconomic security of families in Bradford, data was collected on the ability of families to pay for household bills and food security. A number of categories within other explanatory variables were collapsed to support the analysis owing to small sample sizes within survey responses. These included: general health: satisfactory (comprising of 'excellent', 'very good' and 'good') and unsatisfactory (comprising of 'fair' and 'poor'); food insecurity; secure (comprising of 'never true' or 'sometimes true' that food didn't last) and insecure (comprising of 'often true' that food didn't last); balanced meals: secure (comprising of 'never true' or 'sometimes true' that the household couldn't afford to eat balanced meals) and insecure (comprising of 'often true' that the household couldn't afford to eat balanced meals); housing security: secure (comprising: 'strongly disagree', 'disagree' and 'neither disagree or agree' that I worry about being evicted or having my home repossessed) and insecure (comprising of 'strongly agree' or 'agree' that I worry about being evicted or having my home repossessed); mortgage and rental security: secure (comprising of 'strongly disagree', 'disagree' and 'neither disagree or agree' that I worry about paying for the rent or mortgage) and insecure (comprising of 'strongly agree' or 'agree' that I worry about paying for the rent or mortgage); household bills security: secure (comprising of 'strongly disagree', 'disagree' and 'neither disagree or agree' that I am up to date with household bills) and insecure (comprising of 'strongly agree' or 'agree'); residential status: homeowner (comprising of 'own it outright', 'buying it with the help of a mortgage', and 'part own and part rent/shared ownership') and not homeowner (comprising of 'rent it', 'live here rent free' and 'squatting'). Index of Multiple Deprivation (IMD) decile categories were collapsed into quintiles.

## Data analysis

Descriptive statistics are presented for each of the survey domains. Descriptive analyses of risk of financial insecurity were conducted at the pre-COVID-19 timepoint and at each survey timepoint during the pandemic. Population risk of financial insecurity was examined over time.

Longitudinal multi-level logistic regression models were used, clustered at the level of the individual, to explore differences in financial insecurity by key explanatory variables to explore predictors of financial insecurity over time during the pandemic.

Separate longitudinal multi-level logistic regression models, clustered at the level of the individual, were also conducted to explore whether changes in financial security over time are associated with maternal health and wellbeing outcomes and socioeconomic insecurity.

In order to explore whether or not the magnitude of the association between exposure variables differed between ethnic groups, the multi-level regression models were repeated separately for each ethnic group. This approach avoids the difficulties inherent in interpreting the ethnicity coefficient in regression models controlling for other variables [39]. Missing data on measures was small for most variables and was not adjusted for in the analyses. All statistical analyses were carried out using Stata 15 [40].

## Results

### Study population

Overall, 2043 mothers participated in the phase one survey (administered between 10th April and 30th June 2020); 730 mothers participated in the phase two survey (administered between 29th October and 23rd December 2020); and 903 mothers participated in the phase three survey

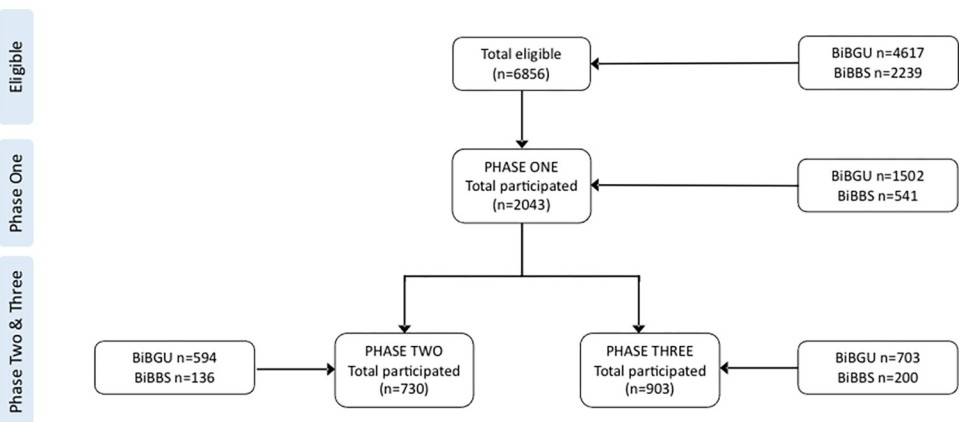

**Fig 1. CONSORT diagram.**

(administered between May and July 2021), see Fig 1. Table 1 describes the number of mothers participating at each phase as a proportion of the eligible population (total BiBBS and BiBGU cohort participants) and phase one survey population by BiBGU and BiBBS cohorts.

Table 2 details the sample baseline characteristics of the phase one study population by ethnic group. Participants were representative of the Bradford population in terms of ethnicity and levels of deprivation [23] and were comparable with regard to key sociodemographics across each phase.

Participants who completed the phase one survey had a mean age of 37 (SD 7) years. Participants were ethnically diverse: 922 (48%) were of Pakistani heritage; 706 (34%) were White British; and 345 (18%) comprised of mothers from other ethnic groups. The majority of participants lived in the first (63%) and second (22%) most deprived IMD quintiles in England. At the onset of the pandemic, most participants reported that the main earner of the household was employed and still working (55%) or was employed and on furlough (15%). Other participants reported that the main earner of the household was self-employed and working (8%), self-employed and not-working (11%) or unemployed (11%). Key worker status was reported by 51% of participants. Most participants owned their own home (66%). Some participants privately (23%) or socially (5%) rented their home. Household composition varied across participants. The average number of adults per household was 2.36 (SD 1.17, range 1–10) with an average of 0.08 (SD 0.31, range 0–3) adults over 70 years old per household. On average, there were 2.52 (SD 1.26, range 0–12) children per household, with an average of 0.63 (SD 0.80, range 0–6) children under the age of 4 years per household. Most participants (78%) reported that they were married. A small proportion of participants reported that they were in a relationship but not married (10%) or were single (12%). 61 (3%) mothers were single parents.

There were several notable differences in key sociodemographics between ethnic groups. Families of Pakistani Heritage (63%) and from other ethnic groups (57%) were significantly more likely to be from the most deprived IMD quintile, compared to White British families (22%), with White British families forming the majority of families from IMD Quintile 3 (77%), IMD Quintile 4 (81%) and IMD Quintile 5 (97%) respectively. Mothers of Pakistani Heritage were significantly more likely to have no qualifications (63%) compared to White British mothers (22%) or mothers of other ethnicity (15%).

The main earner from households where the mother was of 'Other' ethnicity were less likely to be employed and working (18%), compared to White British (41%) and Pakistani Heritage (40%) households and were also less likely to be furloughed (15%), compared to the main

**Table 1. Number of mothers participating at each phase as a proportion of the eligible population (total BiBBS and BiBGU cohort participants) and phase one survey population by BiBGU [24, 25] and BiBBS [26] cohorts.**

| | Eligible population | Phase One | | Phase Two | | | Phase One and Two Complete | | | Phase Three | | | Phase One, Two and Three Complete | | |
|---|---|---|---|---|---|---|---|---|---|---|---|---|---|---|---|
| | | n | Proportion of eligible (%) | n | Proportion of eligible (%) | Proportion of phase one (%) | n | Proportion of eligible (%) | Proportion of phase one (%) | n | Proportion of eligible (%) | Proportion of phase one (%) | n | Proportion of eligible (%) | Proportion of phase one (%) |
| BiBBS | 2239 | 541 | 24 | 136 | 6 | 25 | 80 | 4 | 15 | 200 | 9 | 37 | 46 | 2 | 9 |
| BiBGU | 4617 | 1502 | 33 | 594 | 13 | 40 | 533 | 12 | 35 | 703 | 15 | 47 | 154 | 3 | 10 |
| Total | 6856 | 2043 | 30 | 730 | 11 | 36 | 613 | 9 | 30 | 903 | 13 | 44 | 200 | 3 | 10 |

**Table 2. Sample baseline characteristics of the phase one study population by ethnic group.**

| | Overall (n = 2043)[a] | | White British (n = 706) | | Pakistani Heritage (n = 922) | | Other (n = 345) | |
|---|---|---|---|---|---|---|---|---|
| | Number | Frequency (%) (95% CI) | Number | Frequency (%) (95% CI) | Number | Frequency (%) (95% CI) | Number | Frequency (%) (95% CI) |
| **Index of Multiple Deprivation (IMD) 2019 quintile** | | | | | | | | |
| IMD 1 (most deprived) | 1286 | 63 (61, 65) | 274 | 22 (19, 24) | 713 | 57 (55, 60) | 254 | 20 (19, 23) |
| IMD 2 | 442 | 22 (20, 24) | 196 | 45 (40, 50) | 175 | 40 (36, 45) | 64 | 15 (12, 18) |
| IMD 3 | 155 | 8 (7,9) | 119 | 77 (70, 83) | 20 | 13 (9, 19) | 15 | 10 (6, 16) |
| IMD 4 | 111 | 5 (5, 7) | 87 | 81 (73, 88) | 9 | 8 (4, 15) | 11 | 10 (6, 18) |
| IMD 5 (least deprived) | 34 | 2 (1, 2) | 30 | 97 (80, 100) | <5 | | <5 | |
| Missing | 15 | | <5 | | <5 | | <5 | |
| **Educational status** | | | | | | | | |
| No qualifications | 367 | 19 (17, 21) | 80 | 22 (18, 26) | 230 | 63 (58, 68) | 55 | 15 (12, 19) |
| 5 or fewer GCSEs (grade A-C) or equivalent | 432 | 22 (21, 24) | 181 | 42 (47, 47) | 207 | 48 (43, 53) | 44 | 10 (8, 13) |
| 5 or more GCSEs (grade A-C) or equivalent | 316 | 16 (15, 18) | 124 | 39 (34, 45) | 134 | 42 (37, 48) | 58 | 18 (14, 23) |
| A Levels or equivalent | 680 | 35 (33, 37) | 232 | 34 (36, 54) | 321 | 47 (43, 51) | 127 | 19 (16, 22) |
| Degree or equivalent | 121 | 6 (5, 7) | 54 | 45 (36, 54) | 30 | 31 (23, 39) | 30 | 25 (18, 33) |
| Missing | 127 | | 35 | | <5 | | 31 | |
| **Employment status of main earner** | | | | | | | | |
| Employed: working | 1085 | 55 (52,57) | 436 | 41 (38, 44) | 429 | 40 (38, 43) | 195 | 18 (16, 21) |
| Employed: on furlough | 292 | 15 (13–16) | 107 | 38 (33, 44) | 132 | 47 (41, 53) | 41 | 15 (16, 21) |
| Self-employed: working | 163 | 8 (7, 9) | 44 | 29 (22, 36) | 84 | 55 (47, 62) | 26 | 17 (12, 24) |
| Self-employed: not working | 228 | 11 (10, 13) | 30 | 13 (10, 19) | 164 | 73 (67, 79) | 30 | 13 (10, 19) |
| Unemployed | 221 | 11 (10, 13) | 81 | 38 (32, 45) | 90 | 42 (35, 49) | 43 | 20 (15, 26) |
| Missing | 54 | | 8 | | 23 | | 10 | |
| **Whether mother is a key worker** | | | | | | | | |
| No | 1000 | 49 (47, 52) | 261 | 27 (24, 30) | 531 | 55 (52, 58) | 171 | 18 (16, 20) |
| Yes | 1025 | 51 (48, 53) | 444 | 44 (41, 47) | 384 | 38 (35, 41) | 174 | 18 (15, 20) |
| Missing | 18 | | <5 | | 7 | | <5 | |
| **Whether mother is pregnant** | | | | | | | | |
| No | 1964 | 97 (96, 97) | 693 | 36 (34, 39) | 880 | 46 (44, 49) | 330 | 17 (16, 19) |
| Yes | 71 | 3 (3, 4) | 12 | 17 (10, 27) | 42 | 61 (49, 71) | 15 | 23 (14, 34) |
| Missing | 8 | | <5 | | <5 | | <5 | |
| **Homeownership status** | | | | | | | | |
| Owner occupied | 1341 | 66 (64, 68) | 464 | 35 (32, 37) | 671 | 50 (48, 53) | 196 | 15 (13, 17) |
| Private rental | 455 | 23 (21, 24) | 198 | 42 (38, 47) | 145 | 32 (28, 36) | 110 | 26 (22, 29) |
| Social rental | 94 | 5 (4, 6) | 15 | 16 (10, 24) | 67 | 72 (62, 79) | 12 | 13 (8, 21) |
| Other | 131 | 7 (6, 8) | <5 | | <5 | | <5 | |
| Missing | 22 | | 27 | | 36 | | 26 | |
| **Total adults in household (n)** | | | | | | | | |
| Fewer than 2 | 288 | 15 (14, 17) | 129 | 45 (39, 51) | 99 | 35 (29, 40) | 58 | 20 (16, 25) |
| 2 or more | 1649 | 85 (83, 86) | 567 | 35 (33, 38) | 763 | 47 (45, 50) | 280 | 17 (16, 19) |
| Missing | 106 | | 10 | | 60 | | 7 | |
| **Total adults over 70 years old in household (n)** | | | | | | | | |
| Fewer than 2 | 1867 | 99 (98, 99) | 684 | 38 (35, 40) | 817 | 45 (43, 47) | 316 | 17 (16, 19) |
| 2 or more | 26 | 1 (1,2) | <5 | | 20 | 74 (55, 87) | 6 | 26 (13, 45) |
| Missing | 150 | | 22 | | 85 | | 23 | |
| **Total children in household (n)** | | | | | | | | |

*(Continued)*

**Table 2.** (Continued)

| | Overall (n = 2043)[a] | | White British (n = 706) | | Pakistani Heritage (n = 922) | | Other (n = 345) | |
|---|---|---|---|---|---|---|---|---|
| | Number | Frequency (%) (95% CI) | Number | Frequency (%) (95% CI) | Number | Frequency (%) (95% CI) | Number | Frequency (%) (95% CI) |
| Fewer than 2 | 322 | 19 (17, 21) | 140 | 44 (39, 50) | 116 | 37 (32, 42) | 59 | 19 (15, 23) |
| 2 or more | 1367 | 81 (79, 83) | 465 | 35 (32, 38) | 641 | 48 (46, 51) | 222 | 17 (15, 19) |
| Missing | 354 | | 101 | | 165 | | 64 | |
| **Total children under 4 years old in household (n)** | | | | | | | | |
| Fewer than 2 | 1592 | 86 (85, 88) | 593 | 38 (36, 41) | 685 | 44 (42, 47) | 268 | 17 (16, 19) |
| 2 or more | 249 | 14 (12, 15) | 43 | 18 (13, 23) | 152 | 63 (57, 69) | 47 | 19 (15, 25) |
| Missing | 202 | | 70 | | 85 | | 30 | |
| **People per bedroom (n)** | | | | | | | | |
| Fewer than 2 | 1540 | 81 (79, 83) | 596 | 42 (39, 44) | 508 | 42 (39, 44) | 202 | 16 (14, 18) |
| 2 or more | 364 | 19 (17, 21) | 53 | 17 (14, 22) | 166 | 59 (54, 64) | 67 | 24 (19, 28) |
| Missing | 139 | | 57 | | 248 | | 76 | |
| **Current relationship status** | | | | | | | | |
| Single | 243 | 12 (11, 14) | 138 | 58 (52, 65) | 53 | 22 (18, 28) | 45 | 19 (15, 25) |
| Married | 1571 | 78 (76, 80) | 406 | 27 (24, 29) | 852 | 56 (53, 58) | 270 | 18 (16, 20) |
| In a relationship | 199 | 10 (9, 11) | 158 | 82 (76, 87) | <5 | | 30 | 16 (11, 21) |
| Missing | 30 | | <5 | | 13 | | <5 | |
| **Whether single parent** | | | | | | | | |
| Yes | 61 | 3 (2, 4) | 46 | 78 (66, 87) | 7 | 12 (6, 23) | 6 | 10 (5, 21) |
| No | 1940 | 97 (96, 98) | 656 | 35 (33, 37) | 895 | 47 (45, 50) | 336 | 18 (16, 19) |
| Missing | 42 | | <5 | | 20 | | <5 | |
| **Baseline PHQ-8 category** | | | | | | | | |
| None | 1123 | 57 (55, 59) | 370 | 33 (30, 36) | 560 | 49 (46, 52) | 193 | 18 (16, 20) |
| Mild | 466 | 24 (22, 26) | 141 | 34 (29, 38) | 204 | 49 (44, 54) | 73 | 17 (14, 21) |
| Moderate | 213 | 11 (10, 12) | 43 | 31 (24, 40) | 70 | 51 (43, 59) | 24 | 18 (12, 25) |
| Moderately severe/severe | 159 | 8 (7, 9) | 34 | 47 (34, 59) | 39 | 50 (37, 63) | 6 | 3 (1, 13) |
| Missing | 82 | | 118 | | 49 | | 49 | |
| **Baseline GAD-7 category** | | | | | | | | |
| None | 1206 | 61 (59, 63) | 402 | 33 (31, 36) | 601 | 49 (46, 51) | 203 | 18 (16, 20) |
| Mild | 444 | 23 (21, 24) | 107 | 40 (34, 46) | 122 | 45 (39, 51) | 40 | 15 (11, 20) |
| Moderate | 183 | 9 (8, 11) | 38 | 37 (28, 47) | 52 | 50 (41, 60) | 13 | 13 (7, 21) |
| Severe | 135 | 7 (6, 8) | 22 | 32 (22, 44) | 38 | 56 (44, 67) | 8 | 12 (6, 22) |
| Missing | 75 | | 137 | | 109 | | 81 | |
| **Self-reported general health** | | | | | | | | |
| Excellent | 197 | 10 (9, 11) | 57 | 30 (24, 37) | 94 | 50 (43, 57) | 36 | 19 (14, 26) |
| Very good | 455 | 23 (21, 24) | 199 | 45 (41, 50) | 159 | 36 (32, 41) | 80 | 18 (15, 22) |
| Good | 814 | 40 (38, 43) | 262 | 33 (30, 36) | 393 | 50 (46, 53) | 138 | 17 (15, 20) |
| Fair | 412 | 20 (19, 22) | 141 | 35 (30, 40) | 195 | 48 (43, 53) | 68 | 17 (13, 21) |
| Poor | 135 | 7 (6, 8) | 43 | 32 (25, 40) | 74 | 55 (46, 63) | 18 | 13 (9, 20) |
| Missing | 30 | | <5 | | 7 | | <5 | |
| **Whether anyone in household is clinically vulnerable** | | | | | | | | |
| No | 1567 | 77 (75, 79) | 591 | 39 (37, 41) | 656 | 43 (41, 46) | 269 | 18 (16, 20) |
| Yes | 464 | 23 (21, 25) | 114 | 25 (21, 29) | 265 | 58 (53, 62) | 76 | 17 (14, 21) |
| Missing | 12 | | <5 | | <5 | | <5 | |
| **Whether anyone in household has self-isolated** | | | | | | | | |
| No | 1471 | 73 (71, 74) | 545 | 37 (35, 39) | 673 | 45 (43, 47) | 253 | 18 (16, 20) |

*(Continued)*

**Table 2.** (Continued)

| | Overall (n = 2043)[a] | | White British (n = 706) | | Pakistani Heritage (n = 922) | | Other (n = 345) | |
|---|---|---|---|---|---|---|---|---|
| | Number | Frequency (%) (95% CI) | Number | Frequency (%) (95% CI) | Number | Frequency (%) (95% CI) | Number | Frequency (%) (95% CI) |
| Yes | 558 | 28 (26, 30) | 100 | 29 (25, 34) | 188 | 55 (50, 60) | 51 | 16 (12, 20) |
| Missing | 14 | | 61 | | 61 | | 41 | |

[a] Including 70 missing from ethnicity variable.

earner from White British (38%) and Pakistani Heritage (47%) families. The main earner in families of Pakistani Heritage were notably more likely to be self-employed and not working (73%) compared to White British (13%) and other (13%) households. Overall, there were more White British mothers who were key workers (44%) compared to mothers of Pakistani Heritage (38%) or mothers from other ethnic groups (18%).

Families of Pakistani Heritage were more likely to own their own homes (50%) compared to White British families (35%) and families of other ethnicity (15%) and were more likely to be socially renting (72%) compared to other families. Pakistani Heritage families were more likely to have greater numbers of adults and children in the household compared to White British families or families from other ethnic groups, who were most likely to have two adults and two children per household. Similarly, families of Pakistani Heritage were more likely to have more people per bedroom (59%) than White British families (17%) or families from other ethnic groups (24%), who were most likely to have 2 or 3 people per bedroom.

Mothers of Pakistani Heritage were more commonly married (56%). The majority of mothers who were single (58%) or who were in a relationship and not married (82%) were White British. The majority of single mothers were also White British (78%). Mothers of Pakistani Heritage were more likely to report clinical depression and anxiety, alongside unsatisfactory self-reported general health compared to White British families. Table 1 describes the sample baseline characteristics of the phase one study population by ethnic group.

## Impact of COVID-19 pandemic on financial insecurity

Financial insecurity was frequently reported by families in Bradford throughout the pandemic, most notably in phase one, with the proportion of families reporting financial insecurity having improved but not completely returned to baseline pre-pandemic levels by phase three. Table 3 describes the levels of financial insecurity at pre-pandemic baseline and each COVID-19 survey timepoint. The largest reduction in financial insecurity occurred for families who reported that they were living comfortably, which almost recovered to baseline pre-pandemic proportions by phase three. The proportion of families reporting that they were doing alright remained fairly constant throughout the pandemic. The proportion of families just about getting by did not return to baseline pre-pandemic levels by phase three. Overall, more families reported that they were finding it quite or very difficult financially at phase one (12%) compared to baseline (7%). This improved in phase two (9%) and almost returned to baseline by phase three (8%).

The risk of being financially insecure was greatest in phase one (OR 2.33, 95% CI 1.79, 3.05), with the risk of experiencing financial insecurity almost returning to pre-COVID-19 baseline levels by phase three with the risk no longer statistically significant at phase three, see Table 4.

Overall, the probability of being financially insecure for families in Bradford at phase one was 12.05% (95% CI 10.59, 13.50,), at phase two was 10.20% (95% CI 7.99, 12.41,) and at phase

**Table 3. Levels of financial security at pre-COVID-19 baseline and COVID-19 surveys.**

| Financial security | Pre-COVID-19 baseline | | COVID-19 phase one | | COVID-19 phase two | | COVID-19 phase three | |
|---|---|---|---|---|---|---|---|---|
| | n | Frequency (%) (95% CI) | n | Frequency (%) (95% CI) | n | Frequency (%) (95% CI) | n | Frequency (%) (95% CI) |
| Living comfortably | 684 | 32 (30, 35) | 403 | 20 (18, 22) | 190 | 26 (23, 29) | 257 | 29 (26, 32) |
| Doing alright | 852 | 40 (38, 42) | 857 | 42 (40, 44) | 290 | 40 (36, 44) | 339 | 39 (35, 42) |
| Just about getting by | 410 | 19 (18, 21) | 501 | 25 (23, 27) | 167 | 23 (20, 26) | 197 | 22 (20, 25) |
| Finding it quite difficult | 100 | 5 (4, 6) | 180 | 9 (8, 10) | 51 | 7 (5, 9) | 46 | 5 (4, 7) |
| Finding it very difficult | 42 | 2 (1, 3) | 57 | 3 (2, 4) | 16 | 2 (1, 4) | 24 | 3 (2, 4) |
| Missing | 34 | | 45 | | 16 | | 40 | |
| Total | 2122 | 100 | 2043 | 100 | 730 | 100 | 903 | 100.00 |

three was 8.49% (95% CI 6.66, 10.32), compared to baseline probabilities of 6.96% (95% CI 5.86, 8.07), see Fig 2.

## Individual factors associated with changes to financial security

Several individual characteristics were found to be associated with financial insecurity during the pandemic for mothers and their families. Table 5 describes the probability of experiencing financial insecurity for mothers overall throughout the pandemic and between pre-COVID-19 baseline and COVID-19 lockdown surveys for individual sociodemographics.

**Sociodemographics factors.** Sociodemographic factors, including IMD quintile, employment, and household factors, such as homeowner status and household density, demonstrated expected relationships with financial insecurity. The risk of financial insecurity increased with decreasing IMD quintiles. The risk of being financially insecure decreased overall with greater levels of educational attainment, being statistically significant for those with 5 or more GCSE's (grades A-C) or equivalent (OR 0.38, 95% CI 0.22, 0.67) and A Levels (OR 0.38, 95% CI 0.24, 0.59). The most protective employment status was being employed and working at phase one. The risk was greatest for families where the main earner was unemployed (OR 9.81 95% CI 4.82, 19.95), employed and on furlough (OR 5.30 95% CI 2.77, 10.14) and self-employed and not working (OR 4.59 95% CI 2.64, 7.98) respectively compared to being employed and working. Being a key worker was also a protective factor against financial insecurity (OR 0.49 95% CI 0.36, 0.67). Families who did not own their own homes were also more likely to report financial insecurity overall (OR 2.72 95% CI 1.98, 3.72). Families with two or more adults in the household were less likely to report financial insecurities (OR 0.61 95% CI 0.41, 0.91). Families eligible for free school meals were more likely to report financial insecurities (OR 2.59

**Table 4. Odds ratios (95% CI) from unadjusted mixed-effects logistic regression model for a change in financial security between pre-COVID-19 and COVID-19 pandemic surveys.**

| | OR | p-value | 95% CI |
|---|---|---|---|
| Financial security (Reference: Pre-COVID-19 baseline financial security) | | | |
| Phase One | 2.33 | 0.000 | 1.79–3.03 |
| Phase Two | 1.79 | 0.003 | 1.22–2.61 |
| Phase Three | 1.34 | 0.114 | 0.93–1.94 |

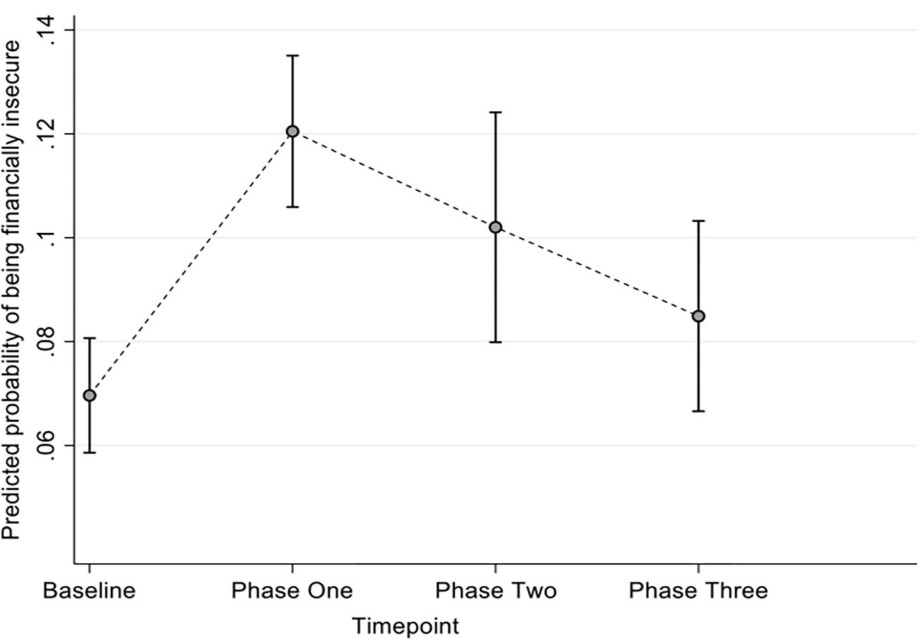

**Fig 2. Marginal predicted means of being financially insecure at baseline, phase one, phase two and phase three surveys.**

95% CI 1.69, 3.97). These associations largely persisted throughout the pandemic but were difficult to examine in detail at each timepoint owing to small sample sizes across strata.

Overall, there was no difference observed in the risk of experiencing financial insecurity for the following variables: age; pregnancy status; families with a child with special educational needs; single parents; by numbers of people per bedroom; number of adults over 70 years old and numbers of children in the household.

**Ethnicity.**   Families were more likely to report being financially insecure if they identified as Pakistani Heritage (OR 2.94, 95% CI 2.03, 4.25) or from other ethnic groups (OR 1.84, 95% CI 1.14, 2.97), compared to White British families.

Over the course of the pandemic, Pakistani Heritage families experienced the sharpest rise in the risk of experiencing financial insecurity at the onset of the pandemic, compared to those of White British families and families from other ethnic groups, see Fig 3. Over the course of the pandemic, the risk of experiencing financial insecurity declined, however had not returned to pre-COVID-19 baseline levels for those of Pakistani Heritage by phase three. Families from other ethnic groups also experienced a rise in probability of financial insecurity in phase one which remained constant across phases two and three, indicating that recovery in this group may also not have occurred. This trend is not statistically significant, likely owing to the smaller sample sizes in this ethnic group. Like families of Pakistani Heritage and other ethnic groups, White British families experienced an increased probability of experiencing financial insecurity during phase one, however this returned to baseline by phase two and appears to have remained at similar levels to baseline at phase three.

After controlling for employment status, being a key worker, IMD quintile, household composition, homeownership status, self-reported general health and baseline PHQ-8 and GAD-7 categories, there remained an association between experiencing financial insecurity and being of Pakistani Heritage (OR 2.22, 95% CI 1.43, 3.45) compared to families of White British

**Table 5. Odds ratios (95% CI) from unadjusted mixed-effects logistic regression models for the probability of experiencing financial insecurity for participants overall across the pandemic and between pre-COVID-19 baseline and COVID-19 lockdown surveys for individual sociodemographic characteristics.**

| | Overall | | Phase one | | Phase two | | Phase three | |
|---|---|---|---|---|---|---|---|---|
| | OR | 95% CI | OR | 95% CI | OR | 95% CI | OR | 95% CI |
| **Ethnicity (Reference: White British)** | | | | | | | | |
| Pakistani Heritage | 2.94 | 2.03–4.25 | 2.62 | 1.41–4.85 | 2.82 | 1.14–6.96 | 1.83 | 0.77–4.33 |
| Other | 1.84 | 1.14–2.97 | 1.01 | 0.45–2.27 | 1.30 | 0.39–4.33 | 1.340 | 0.45–3.97 |
| **Age on 1st April 2020 (Reference: 18–30 years)** | | | | | | | | |
| 30–34 years | 1.21 | 0.51–2.88 | 2.05 | 0.56–7.49 | 0.87 | 0.08–9.18 | - | - |
| 35–39 years | 1.02 | 0.44–2.40 | 1.30 | 0.36–4.65 | 0.92 | 0.09–9.18 | 1.61 | 0.59–4.38 |
| 40–44 years | 0.77 | 0.31–1.96 | 1.11 | 0.28–4.44 | 0.23 | 0.02–2.96 | 0.70 | 0.26–1.85 |
| Over 45 years | 4.38 | 0.16–120.04 | 0.85 | 0.02–31.33 | - | - | - | - |
| **IMD 2019 quintile (Reference: IMD 1 (most deprived))** | | | | | | | | |
| IMD 2 | 0.37 | 0.25–0.56 | 1.01 | 0.52–1.97 | 1.45 | 0.59–3.58 | 0.84 | 0.30–2.34 |
| IMD 3 | 0.15 | 0.07–0.32 | 0.81 | 0.22–3.01 | 1.41 | 0.29–6.84 | 0.46 | 0.05–4.34 |
| IMD 4 | 0.12 | 0.04–0.31 | 0.10 | 0.01–0.89 | - | - | 0.63 | 0.11–3.49 |
| IMD 5 (least deprived) | 0.09 | 0.01–0.54 | 0.55 | 0.03–9.35 | - | - | - | - |
| **Educational status (Reference: No qualifications)** | | | | | | | | |
| 5 or fewer GCSE (grades A-C) or equivalent | 0.65 | 0.40–1.04 | 0.72 | 0.36–1.41 | 0.38 | 0.08–1.82 | 0.75 | 0.23–2.49 |
| 5 or more GCSE's (grades A-C) or equivalent | 0.38 | 0.22–0.67 | 0.80 | 0.35–1.84 | 1.71 | 0.44–6.62 | 1.49 | 0.43–5.12 |
| A Levels or equivalent | 0.38 | 0.24–0.59 | 0.92 | 0.47–1.80 | 2.23 | 0.71–7.01 | 1.13 | 0.38–3.34 |
| Degree or equivalent | 0.89 | 0.46–1.76 | 0.60 | 0.22–1.68 | 2.01 | 0.44–9.09 | 1.48 | 0.36–6.16 |
| **Employment status of main earner (Reference: Employed: working)** | | | | | | | | |
| Employed: on furlough | 5.30 | 2.77–10.14 | 1.28 | 0.06–28.98 | 0.91 | 0.03–29.20 | - | - |
| Self-employed: working | 2.09 | 0.86–5.03 | 0.20 | 0.02–2.71 | 1.12 | 0.04–29.15 | - | - |
| Self-employed: not working | 4.59 | 2.64–7.98 | 5.42 | 1.74–16.87 | 1.00 | 0.25–4.02 | - | - |
| Unemployed | 9.81 | 4.82–19.95 | 2.98 | 0.13–70.79 | 23.60 | 0.79–707.26 | - | - |
| **Whether mother is a key worker (Reference: Not key worker)** | | | | | | | | |
| Key worker | 0.49 | 0.36–0.67 | 0.48 | 0.28–0.82 | 0.42 | 0.19–0.91 | 0.89 | 0.43–1.84 |
| **Whether mother is pregnant (Reference: Not pregnant)** | | | | | | | | |
| Pregnant | 1.04 | 0.44–2.47 | 0.81 | 0.21–3.20 | 2.33 | 0.18–30.24 | - | - |
| **Homeownership status (Reference: Homeowners)** | | | | | | | | |
| Non-homeowners | 2.72 | 1.98–3.72 | 1.00 | 0.59–1.71 | 1.35 | 0.61–2.96 | 1.02 | 0.48–2.16 |
| **Total adults in household (Reference: Fewer than 2)** | | | | | | | | |
| 2 or more | 0.61 | 0.41–0.91 | 1.80 | 0.91–3.55 | 1.27 | 0.38–4.17 | 0.94 | 0.25–3.53 |
| **Total adults in household over 70 years old (Reference: Fewer than 2)** | | | | | | | | |
| 2 or more | 0.36 | 0.06–2.24 | 0.17 | 0.01–6.12 | - | - | - | - |
| **Total children in household (Reference: Fewer than 2)** | | | | | | | | |
| 2 or more | 1.13 | 0.76–1.69 | 1.00 | 0.49–2.06 | 0.60 | 0.19–1.86 | 0.43 | 0.12–1.50 |
| **Total children under 4 years old in household (Reference: Fewer than 2)** | | | | | | | | |
| 2 or more | 1.05 | 0.66–1.67 | 1.46 | 0.65–3.26 | 0.85 | 0.14–5.29 | 0.71 | 0.15–3.43 |
| **Number of people per bedroom (Reference: Fewer than 2)** | | | | | | | | |
| 2 or more | 1.36 | 0.92–1.99 | 1.49 | 0.28–3.96 | 1.35 | 0.25–7.18 | - | - |

(*Continued*)

**Table 5.** (Continued)

| | Overall | | Phase one | | Phase two | | Phase three | |
|---|---|---|---|---|---|---|---|---|
| | OR | 95% CI | OR | 95% CI | OR | 95% CI | OR | 95% CI |
| **Free school meal (Reference: Not eligible)** | | | | | | | | |
| Eligible | 2.59 | 1.69–3.97 | 0.77 | 0.39–1.52 | 0.92 | 0.33–2.53 | 0.97 | 0.35–2.67 |
| **Special educational needs (Reference: No special educational needs)** | | | | | | | | |
| Special educational needs | 0.98 | 0.61–1.58 | 2.98 | 1.23–7.24 | 2.48 | 0.76–8.13 | 1.66 | 0.43–6.48 |
| **Whether single parent (Reference: Not a single parent)** | | | | | | | | |
| Single parent | 0.40 | 0.14–1.12 | 0.36 | 0.04–3.41 | 2.09 | 0.24–18.41 | 2.96 | 0.27–32.69 |
| **PHQ-8 Category at baseline (Reference: No depression)** | | | | | | | | |
| Mild depression | 1.77 | 1.21–2.59 | 0.75 | 0.40–1.39 | 0.55 | 0.19–1.57 | 0.76 | 0.30–1.94 |
| Moderate depression | 6.70 | 3.98–11.30 | 0.47 | 0.22–0.99 | 1.83 | 0.67–4.97 | 1.34 | 0.46–3.88 |
| Severe depression | 6.43 | 3.34–12.39 | 0.37 | 0.16–0.89 | 0.23 | 0.05–1.45 | 0.26 | 0.05–1.35 |
| **GAD-7 Category at baseline (Reference: No anxiety)** | | | | | | | | |
| Mild anxiety | 4.48 | 2.36–8.48 | 0.52 | 0.27–1.01 | 0.41 | 0.15–1.14 | 1.00 | 0.39–2.59 |
| Moderate anxiety | 8.85 | 3.84–20.39 | 0.46 | 0.20–1.06 | 1.05 | 0.32–3.47 | 1.05 | 0.30–3.61 |
| Severe anxiety | 16.73 | 6.56–42.66 | 0.32 | 0.13–0.79 | 0.21 | 0.04–1.13 | 0.52 | 0.12–2.24 |
| **Whether anyone in household is clinically vulnerable to COVID-19 (Reference: Not clinically vulnerable)** | | | | | | | | |
| Clinically vulnerable | 1.12 | 0.78–1.61 | 2.54 | 1.33–4.85 | 1.66 | 0.65–4.22 | 1.76 | 0.71–4.36 |

Missing data omitted due to collinearity indicated by–symbol.

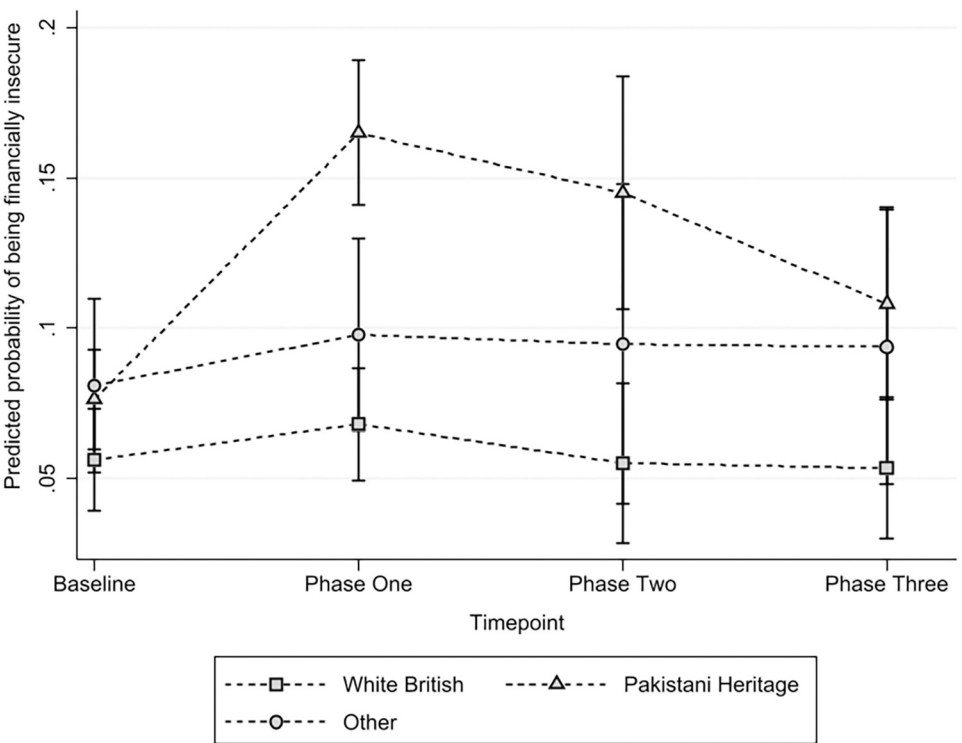

**Fig 3. Marginal predicted means of financial insecurity reported by ethnic group across the pandemic.**

heritage. This indicates that despite differences in baseline characteristics between these ethnic groups, being of Pakistani Heritage is an independent risk factor for financial insecurity throughout the pandemic.

## Impact of financial insecurity on maternal health, wellbeing and socioeconomic insecurity

Table 6 describes the association between financial insecurity and health, wellbeing and socio-economic security experienced by mothers and their families throughout the pandemic and at individual timepoints.

**Self-reported general health.** Overall mothers with unsatisfactory self-reported general health were more likely to report being financially insecure throughout the pandemic (OR 3.36, 95% CI 2.45, 4.62) compared to those who reported satisfactory health. Overall there was no association found between financial insecurity and households with a family member who is clinically vulnerable to COVID-19 (OR 1.12 95% CI 0.78, 1.61). However, at phase one,

**Table 6. Odds ratios (95% CI) from unadjusted mixed-effects logistic regression models and unadjusted logistic regression models for a change in financial security between pre-COVID-19 and COVID-19 lockdown surveys.**

| | Overall | | Phase one | | Phase two | | Phase three | |
|---|---|---|---|---|---|---|---|---|
| | OR | 95% CI | OR | 95% CI | OR | 95% CI | OR | 95% CI |
| Self-reported general health (Reference: Satisfactory)** | | | | | | | | |
| Unsatisfactory | 3.36 | 2.45–4.62 | | | | | | |
| **PHQ-8 Category (Reference: No depression at baseline)** | | | | | | | | |
| Mild depression | 2.80 | 2.01–3.89 | 1.30 | 0.61–2.76 | 3.04 | 1.08–8.58 | 1.65 | 0.59–4.58 |
| Moderate depression | 4.87 | 3.22–7.35 | 0.70 | 0.28–1.78 | 1.37 | 0.38–4.93 | 0.92 | 0.27–3.18 |
| Severe depression | 13.79 | 8.71–21.85 | 1.04 | 0.39–2.75 | 1.08 | 0.28–4.18 | 1.83 | 0.51–6.55 |
| **Clinical depression (Reference: No clinical depression)** | | | | | | | | |
| Clinical depression | 5.27 | 3.75–7.40 | 0.84 | 0.42–1.65 | 0.59 | 0.19–1.80 | 1.20 | 0.34–4.31 |
| **GAD-7 Category (Reference: No anxiety at baseline)** | | | | | | | | |
| Mild anxiety | 3.52 | 2.52–4.91 | 0.91 | 0.42–1.94 | 0.55 | 0.16–1.82 | 1.33 | 0.48–3.68 |
| Moderate anxiety | 6.05 | 3.94–9.23 | 0.83 | 0.31–2.21 | 1.40 | 0.33–5.98 | 1.92 | 0.53–6.86 |
| Severe anxiety | 13.87 | 8.50–22.63 | 0.87 | 0.31–2.42 | 0.78 | 0.16–3.71 | 1.02 | 0.26–3.95 |
| **Clinical anxiety (Reference: No clinical anxiety)** | | | | | | | | |
| Clinical anxiety | 5.70 | 4.01–8.10 | 0.88 | 0.43–1.81 | 0.97 | 0.27–3.54 | 1.05 | 0.30–3.70 |
| **Worry about paying for rent or mortgage (Reference: Not worried at phase one)** | | | | | | | | |
| Worried | 13.48 | 8.91–20.40 | | | 1.25 | 0.54–2.91 | 1.07 | 0.48–2.40 |
| **Worry about eviction (Reference: Not worried at phase one)** | | | | | | | | |
| Worried | 9.47 | 5.88–15.26 | | | 0.91 | 0.30–2.75 | 0.65 | 0.21–1.96 |
| **Ability to pay bills (Reference: Not up to date with bills at phase one)** | | | | | | | | |
| Up to date with bills | 0.07 | 0.04–0.10 | | | 0.53 | 0.21–1.30 | 0.86 | 0.37–1.99 |
| **Whether food lasted (Reference: Food did last at phase one)** | | | | | | | | |
| Food did not last | 21.57 | 14.05–33.11 | | | 0.96 | 0.41–2.26 | 1.18 | 0.52–2.64 |
| **Ability to eat a balanced meal (Reference: Able to eat a balanced meal at phase one)** | | | | | | | | |
| Not able to eat a balanced meal | 23.20 | 14.34–37.53 | | | 1.95 | 0.78–4.85 | 1.44 | 0.62–3.34 |
| **Needing to skip a meal (Reference: Did not need to skip meals at phase one)** | | | | | | | | |
| Needed to skip meals | 34.29 | 16.98–69.22 | | | 0.50 | 0.15–1.68 | 1.39 | 0.40–4.81 |
| **Feeling hungry (Reference: Not hungry at phase one)** | | | | | | | | |
| Hungry | 92.77 | 32.64–263.62 | | | 1.08 | 0.20–5.74 | 2.91 | 0.70–12.11 |

Missing data omitted due to collinearity indicated by–symbol.

these families had a greater risk of experiencing financial insecurity compared to households who did not have a family member who was clinically vulnerable to COVID-19 (OR 2.54 95% CI 1.33, 4.85). After controlling for employment and key worker status, the association between clinical vulnerability and financial insecurity persisted at phase one (OR 2.06 95% CI 1.07, 3.97).

**Mental health.**   Mothers were more likely to report being financially insecure if they suffered with mild (OR 1.77, 95% CI 1.21, 2.59), moderate (OR 6.70, 95% CI 3.98, 11.30) or severe depression (OR 6.43, 95% CI 3.34, 12.39) compared to those with no symptoms or signs of depression before the onset of the pandemic. The difference between the risk of experiencing financial insecurity reported between mothers with moderate and severe depression at baseline was not significant, although the sample size of mothers with severe depression at baseline was lower than that of the other categorical groups.

Similarly, mothers were also more likely to report being financially insecure if they suffered with mild (OR 4.48, 95% CI 2.36, 8.48), moderate (OR 8.85, 95% CI 3.84, 20.39) or severe anxiety (OR 16.73, 95% CI 6.56, 42.66) compared to those with no signs or symptoms of anxiety at baseline, with the probability of financial insecurity increasing with increasing GAD-7 categorical group.

Overall throughout the pandemic, financial insecurity was strongly associated with clinical depression (OR 5.27 95% CI 3.75, 7.40) and clinical anxiety (OR 5.70 95% CI 4.01, 8.10). At each timepoint across the course of the pandemic, financial insecurity was strongly associated with clinical depression, although this was not statistically significant at phase two, see Fig 4. The risk of financial insecurity for those who were clinically depressed was greatest at phase one and three. After controlling for ethnicity, being a key worker, employment status, feeling lonely and feeling worried about paying for the bills, rent, mortgage and being evicted, there

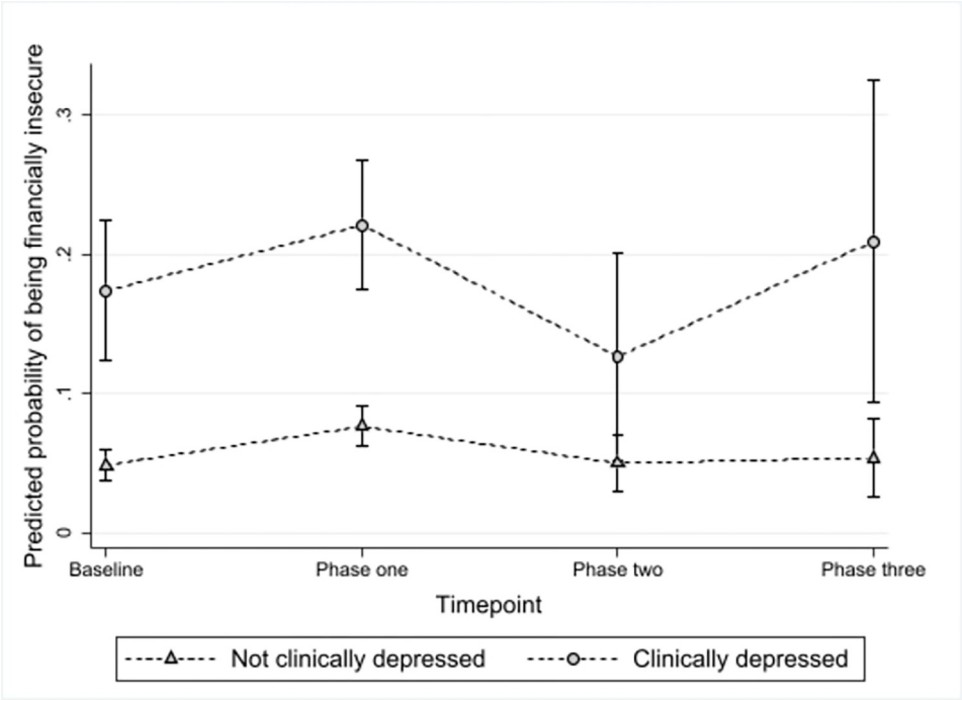

**Fig 4. Marginal predicted means of financial insecurity across the pandemic by clinical depression status, adjusted for ethnicity, being a key worker, employment status, feeling lonely and feeling worried about paying for the bills, rent, mortgage and being evicted.**

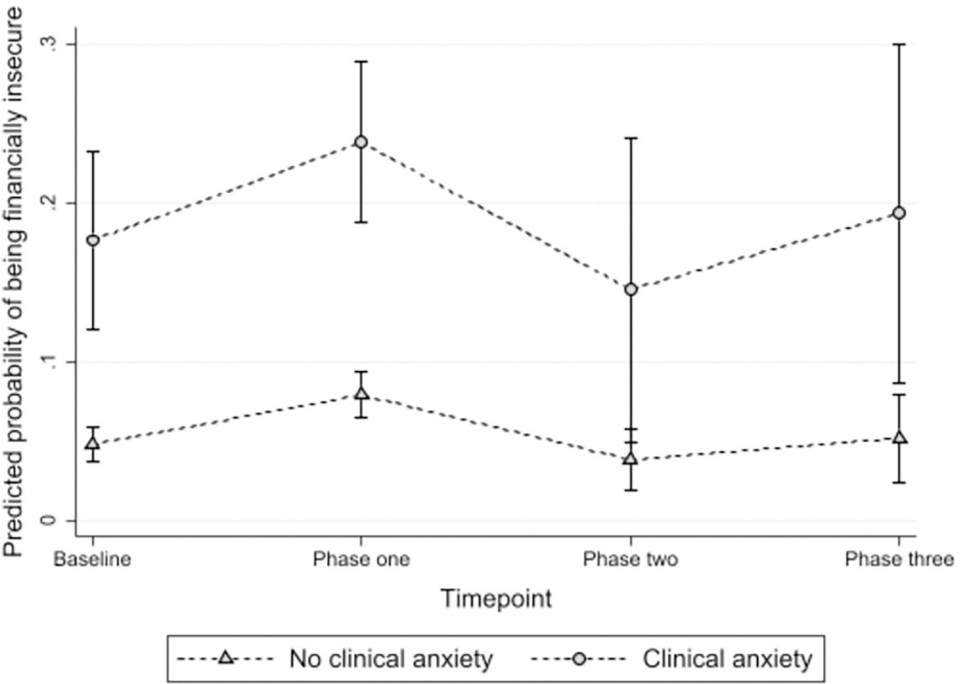

**Fig 5. Marginal predicted means of financial insecurity across the pandemic by clinical anxiety status, adjusted for controlling for ethnicity, being a key worker, employment status, feeling lonely and feeling worried about paying for the bills, rent, mortgage and being evicted.**

remained a strong association between financial insecurity and clinical depression (OR 4.47 95% CI 3.13, 6.39).

At each timepoint across the course of the pandemic, financial insecurity was also strongly associated with clinical anxiety, although this was not statistically significant at phase two, see Fig 5. The risk of financial insecurity for those with clinical anxiety was greatest at phase one and three. After controlling for ethnicity, being a key worker, employment status, feeling lonely and feeling worried about paying for the bills, rent, mortgage and being evicted, there remained a strong association between financial insecurity and clinical anxiety (OR 4.57 95% CI 3.14, 6.66).

**Food security.** For those experiencing financial insecurity, throughout the entirety of the pandemic, there were strong associations with food insecurity, with those experiencing aspects of food insecurity not showing any meaningful improvement throughout the pandemic. Families reporting financial insecurity throughout the pandemic were highly likely to report: food not lasting (OR 21.57 95% CI 14.05, 33.11); not being able to eat a balanced meal (OR 23.20 95% CI 14.34, 37.53); feeling hungry (OR 92.77 95% CI 32.64, 263.62); needing to skip meals (OR 34.29 95% CI 16.98, 69.22) and needing to access food banks (OR 6.52 95% CI 2.53, 16.83).

**Housing security.** Experiencing financial insecurity throughout the pandemic was strongly associated with increased concern and worry about paying for the rent or mortgage (OR 9.47 95% CI 5.88, 15.26) and eviction (OR 13.48 95% CI 8.91, 20.40). Families reporting financial insecurity are also less likely to be up to date with paying their bills (OR 0.07 95% CI 0.04–0.10). Across the pandemic, concerns regarding household security were strongly associated with financial insecurity at each timepoint, with concerns gradually falling across the course of the pandemic.

**Table 7. Overall odds ratios (95% CI) from unadjusted mixed-effects logistic regression model for a change in financial security between pre-COVID-19 and COVID-19 lockdown surveys by ethnic group throughout the pandemic.**

| | White British | | Pakistani Heritage | | Other | |
|---|---|---|---|---|---|---|
| | OR | 95% CI | OR | 95% CI | OR | 95% CI |
| **Self-reported general health (Reference: Satisfactory)** | | | | | | |
| Unsatisfactory | 6.23 | 3.23–12.01 | 2.97 | 1.98–4.47 | 1.31 | 0.52–3.28 |
| **Clinical depression (Reference: No clinical depression)** | | | | | | |
| Clinical depression | 7.67 | 3.94–14.94 | 4.24 | 2.78–6.47 | 6.92 | 2.69–17.81 |
| **Clinical anxiety (Reference: No clinical anxiety)** | | | | | | |
| Clinical anxiety | 8.38 | 4.29–16.41 | 4.28 | 2.71–6.77 | 9.73 | 3.50–27.08 |
| **Worry about paying for rent or mortgage (Reference: Not worried)** | | | | | | |
| Worried | 9.20 | 3.80–22.27 | 15.15 | 6.99–32.84 | 15.85 | 4.70–53.47 |
| **Worry about eviction (Reference: Not worried)** | | | | | | |
| Worried | 4.61 | 1.25–17.00 | 9.41 | 4.19–21.12 | 14.54 | 5.32–39.73 |
| **Ability to pay bills (Reference: Up to date with bills)** | | | | | | |
| Not up to date with bills | 0.05 | 0.02–0.13 | 0.12 | 0.06–0.22 | 0.07 | 0.03–0.18 |
| **Whether food lasted (Reference: Food did last)** | | | | | | |
| Food did not last | 27.94 | 11.43–68.30 | 17.73 | 9.26–33.94 | 32.68 | 7.71–138.61 |
| **Ability to eat a balanced meal (Reference: Able to eat a balanced meal)** | | | | | | |
| Not able to eat a balanced meal | 24.20 | 9.96–58.82 | 19.12 | 9.38–39.00 | 18.55 | 4.83–71.17 |
| **Needing to skip a meal (Reference: Did not need to skip meals)** | | | | | | |
| Needed to skip meals | 49.75 | 16.44–150.60 | 56.41 | 14.13–225.18 | 8.24 | 0.15–461.25 |
| **Feeling hungry (Reference: Not hungry)** | | | | | | |
| Hungry | 80.39 | 20.56–314.42 | 152.23 | 19.18–1208.11 | 132.00 | 5.15–3385.14 |

**Impact by ethnic group.** The impact of financial insecurity on maternal health, wellbeing and socioeconomic insecurity experienced by mothers and their families in Bradford is presented in Table 7. Detailed analysis by ethnic group was limited by sample size and revealed no statistically significant difference between ethnic groups for the impact of financial insecurity on maternal health and wellbeing outcomes and socioeconomic security outcomes.

## Discussion

### Summary of key findings

This longitudinal study, nested within two longitudinal Born in Bradford cohort studies, describes some of the key experiences of families living in the deprived and ethnically diverse city of Bradford during the COVID-19 pandemic. The findings highlight that financial insecurities were frequently reported by families in Bradford throughout the pandemic, most notably in phase one, with the risk of experiencing financial insecurity almost having returned to pre-COVID-19 baseline levels by phase three. Overall, the probability of being financially insecure for families in Bradford at phase one was 12.05% (95% CI 10.59, 13.50), at phase two was 10.20% (95% CI 7.99, 12.41) and at phase three was 8.49% (95% CI 6.66, 10.32), compared to baseline probabilities of 6.96% (95% CI 5.86, 8.07).

The study highlighted that there were several individual sociodemographic characteristics that were potential predictors of financial insecurity throughout the pandemic. Financial insecurity was strongly associated with homeowner status and free school meal eligibility. Several characteristics were identified as protective against financial insecurity: higher IMD Quintiles (i.e. families living in more affluent areas); greater levels of educational attainment; and

families with two or more adults in the household. Several employment factors were also found to be protective. Where the main earner of the household was employed and working during the pandemic was most protective against being financially insecure. The risk was greatest for families where the main earner was unemployed, employed and on furlough and self-employed and not working respectively compared to being employed and working. Being a key worker was also a protective factor against financial insecurity. It was found that there was no difference in the risk of experiencing financial insecurity: according to age; for those who were pregnant; for families with a child with special educational needs; for single parents; by numbers of people per bedroom; or by household composition with respect to number of adults over 70 years old and numbers of children.

Notably, the findings from this study have demonstrated that families of Pakistani Heritage and from other ethnic groups have been disproportionately affected by financial insecurity owing to the COVID-19 pandemic and subsequent public health measures implemented to control the virus. Ethnicity was demonstrated to be an independent predictor of financial insecurity throughout the COVID-19 pandemic, with families of Pakistani Heritage and families from other ethnic groups being more likely to experience financial insecurity than White British families. This association persisted after controlling for employment status, being a key worker, IMD quintile, homeownership status, household composition, self-reported general health and baseline PHQ-8 and GAD-7 categories.

Over the course of the pandemic, Pakistani Heritage families experienced the sharpest rise in risk of financial insecurity at the onset of the pandemic, compared to those of White British families and families from other ethnic groups, and had not returned to baseline levels by phase three. For White British families, recovery from financial insecurity had been achieved by Phase Two. Further studies are required to establish the causal mechanisms through which this association occurs.

This study demonstrates that there were strong associations also demonstrated between financial insecurity and poor maternal health and wellbeing outcomes. Overall, mothers experiencing financially insecurity were more likely to report unsatisfactory general health, clinical depression and clinical anxiety. The association between financial association and clinical anxiety and depression persisted throughout the pandemic with levels of clinical anxiety and depression appearing to recover in phase two, then beginning to rise again in phase three.

Families experiencing financial insecurity throughout the pandemic were also significantly more likely to suffer detrimental impacts to household and food security. Experiencing financial insecurity throughout the pandemic was strongly associated with increased worries about paying for bills and the rent or mortgage, with consequent concerns regarding eviction. Families were also significantly more likely to report: food not lasting; not being able to eat a balanced meal; feeling hungry; needing to skip meals; and needing to access food banks as a result of financial insecurity.

Several studies have since provided further evidence demonstrating the unequal effects of the pandemic on ethnic minority groups, further exacerbating existing inequalities. In recently published studies, ethnic minority groups were found to be more likely to experience economic hardship immediately after the first national lockdown [41, 42] in keeping with the results of this study. Furthermore, people from ethnic minority groups were found to be more likely to experience loss of employment and less likely to receive furlough payments compared to White British populations [41]. Several studies have confirmed that levels of financial insecurity have not yet returned to baseline levels for ethnic minority groups [42, 43]. The findings, together with the findings of this study, provide evidence that the pandemic has exacerbated entrenched socioeconomic inequalities along intersecting ethnic lines [41–43].

Other studies have published evidence supportive of the findings of this study to suggest that mental health and wellbeing worsened throughout the pandemic and was associated with financial insecurity. Solomon-Moore et al. described that impacts on mental health were the greatest for women, people living with young children and those between 18 and 34 years old [44]. Whilst this study did not examine differences in gender, the study did look at women of child-bearing age, many of whom already had children or who were expecting children and describe findings consistent with this study. While there is some research available on how COVID-19 lockdown restrictions have had an impact on mental health for UK adults [45, 46], data are limited, and not enough is known about potential long-term effects of the pandemic. This study expands on this data, and alongside the findings of some other studies, demonstrates that mental health and wellbeing are improving across time, suggesting that any negative effects of the pandemic on mental health may be reversible [44]. This perceived recovery in mental health and wellbeing may have been due to the easing of public health restrictions, which enabled increased freedom to see family and friends, participate in hobbies and allow some individuals to return to work and thereby lessening financial insecurity. However, the results of this study, provides evidence suggestive of another fall in levels of mental health and wellbeing. Further work needs to be conducted to examine this trend over time and if and how this correlates with financial insecurity and the effects of other emerging socioeconomical and political factors.

## Limitations

A wide array of methods were employed in order to maximise survey response rates in a time sensitive manner. However, the overall low response rates to each survey, as a proportion of both the eligible population and of those having completed previous surveys, may have introduced selection bias. Comparing results with other studies of similar and differing populations will be important to gain a fuller picture of the impact of the pandemic and its management on health and social inequalities. Notably, response rates were lower in phase two and three compared to phase one, limiting analysis at the later stages of the pandemic. However, whilst it is possible that the results are influenced by the overall survey response rates, participants were representative of the Bradford population and BiBGU [24, 25] and BiBBS [26] cohorts, were comparable across phases, and have demonstrated a wide variability in most characteristics.

This study reports several significant associations with financial insecurity for mothers and their families in Bradford. It is not possible from this analysis to establish temporality and thus determine causality for these associations. However, the study has highlighted the direction and magnitude of these relationships for this population, emphasising the need to address all health, social and economic factors to support families to recover holistically, with targeted support to those most vulnerable.

A number of variables were also collapsed to support the analysis owing to small sample sizes within each strata across survey timepoints. For example, financial insecurity was defined as those 'finding it quite difficult' and 'finding it very difficult' to manage financially. Families 'living comfortably', 'doing alright' and 'just about getting by' were considered financially secure. Similarly, families were defined as having food security if it was 'never true' or 'sometimes true' that food didn't last and being food insecure if it was 'often true' that food didn't last. Such categorisations were conservative and several mid-point categories could be considered true for either categorisation.

Furthermore, baseline pre-COVID-19 measurements were taken from data collected over the four years preceding the onset of the pandemic, therefore all changes cannot with confidence be attributed to the pandemic and subsequent public health measures implemented.

## Implication of findings

The COVID-19 pandemic had a demonstrable and profound impact on the wellbeing of families in a range of countries. The fear and uncertainty of the health risks, alongside the stress from ensuing restrictions and constraints on everyday life caused major disruptions to the financial, emotional, and physical wellbeing of families [41–43, 47].

Governments across the world introduced extensive labour market and social policy measures with the aim of retaining jobs and protecting the livelihoods of individuals, and to avert the most dramatic economic and social consequences. Across Europe, the vast majority of economic measures adopted were of an ad hoc nature, highly cost-intensive and intended to be in place for a limited duration only [47, 48]. Whilst such measures appeared novel, comparative international analyses of such welfare measures introduced, such as those be Andrade et al., suggest these measures could be classed as social compensation, an established feature of European welfare states [48]. The measures implemented therefore only differed in their scope, rather than their normative and legal basis in the welfare state, and did not inherently change the underlying structures of welfare systems [48].

It is therefore unsurprising that the global body of evidence suggests that the effects of the pandemic have affected families across the globe in a very similar way. The pandemic disproportionally impacted lower-income families, families from ethnic minority and vulnerable groups, and women [41–43, 47, 48].

This study offers a unique assessment of the socioeconomic impact of the pandemic longitudinally in a highly ethnically diverse, seldom studied population, with a pre-pandemic baseline, the majority of whom live in the most deprived quintiles in the UK and are more vulnerable to mental health conditions. Recovery from the effects of the pandemic for all has been further hampered by the emerging cost of living and energy crisis. A recent report from the International Monetary Fund highlights that the energy crisis is currently affecting UK households harder than any country in western Europe, with the difference between the cost burden on poor and rich households being far more unequal in the UK compared with other countries [49]. With ever increasing cost of living, energy prices and inflation since the pandemic, the ability of families to recover from the effects of the pandemic is untenable without intervention.

Furthermore, the potential ethnic differences in the magnitude of the associations between financial insecurity and health, wellbeing and socioeconomic security reported in this study, and supported by findings published by other studies, warrant further investigation, including an understanding of potentially differing risk and protective factors in different ethnic groups.

Having established that the crisis measures adopted by welfare states during the COVID-19 pandemic are not as novel and transformative as may seem at first sight, it is likely that the phaseout of the crisis measures will reveal some of the more underlying welfare state mechanisms: once the extraordinary measures expire, the difference between those who are covered by regular social protection and those who are not will once again become visible.

With the gradual withdrawal of welfare measures adopted during the pandemic, vulnerable families will be exposed to ongoing economic challenges with limited or no financial resilience and threatens to widen existing inequalities. Policy makers and commissioners must intervene to provide greater support to families. Families need support to enable them to manage financially and stop them becoming homeless and living in food and financial poverty. Increasing access to support for health and wellbeing is also critical in the recovery from the pandemic and beyond. There is also a need to develop methods to reassure and encourage vulnerable families to access health and social support services they need with immediate effect to stop these health inequalities from worsening.

## Conclusion

This longitudinal study provides a comprehensive analysis of some of the key and unequal experiences of families living in the deprived and ethnically diverse city of Bradford during the COVID-19 pandemic. This research, with an extended time scope and pre-COVID-19 baseline data, provides a more extensive analysis of the financial and subsequent impacts on health and wellbeing across several social groups. The findings of this study highlight that the impact of financial insecurity experienced by mothers and their families throughout the COVID-19 pandemic was severe, wide ranging and affected the most vulnerable. Although there were indications that severe financial insecurity was recovering towards the end of the pandemic, the emerging cost of living and energy crisis likely means that the recovery from the effects of the pandemic will be short lived and continues to threaten the health, wellbeing and socioeconomic security of vulnerable families and widen existing health inequalities for the most vulnerable. The need for policy makers and commissioners to act to support vulnerable families is now urgent and critical to prevent further financial, fuel and food debt, homelessness, poor health and widening existing health and social inequalities.

## Acknowledgments

Born in Bradford (BiB) is only possible because of the enthusiasm and commitment of the children and parents in BiB. We are grateful to all the participants, parent governors and community research advisory group members, schools, health professionals and researchers who have made BiB happen.

## Author Contributions

**Conceptualization:** Sian Reece, Josie Dickerson, Brian Kelly, Rosemary R. C. McEachan, Kate E. Pickett.

**Data curation:** Brian Kelly, Kate E. Pickett.

**Formal analysis:** Sian Reece, Josie Dickerson, Brian Kelly, Rosemary R. C. McEachan, Kate E. Pickett.

**Funding acquisition:** Josie Dickerson, Rosemary R. C. McEachan, Kate E. Pickett.

**Investigation:** Josie Dickerson, Brian Kelly.

**Methodology:** Sian Reece, Josie Dickerson, Rosemary R. C. McEachan, Kate E. Pickett.

**Project administration:** Rosemary R. C. McEachan.

**Resources:** Rosemary R. C. McEachan.

**Supervision:** Josie Dickerson, Kate E. Pickett.

**Writing – original draft:** Sian Reece.

**Writing – review & editing:** Sian Reece, Josie Dickerson, Brian Kelly, Kate E. Pickett.

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
