## [Decision Letter · Decision Letter 0]

22 May 2023

PONE-D-23-05276The long-term impact of the Covid-19 pandemic on financial insecurity in vulnerable families: findings from the Born in Bradford Covid-19 longitudinal study.PLOS ONE

Dear Dr. Reece,

Thank you for submitting your manuscript to PLOS ONE. After careful consideration, we feel that it has merit but does not fully meet PLOS ONE’s publication criteria as it currently stands. Therefore, we invite you to submit a revised version of the manuscript that addresses the points raised during the review process.

We look forward to receiving your revised manuscript.

Kind regards,

Wajiha Haq, PhD

Academic Editor

PLOS ONE

Journal Requirements:

2. Please ensure that you have specified (1) whether consent was informed and (2) what type you obtained (for instance, written or verbal, and if verbal, how it was documented and witnessed). If your study included minors, state whether you obtained consent from parents or guardians. If the need for consent was waived by the ethics committee, please include this information.

5. Please note that in order to use the direct billing option the corresponding author must be affiliated with the chosen institute. Please either amend your manuscript to change the affiliation or corresponding author, or email us at plosone@plos.org with a request to remove this option.

Reviewers' comments:

Reviewer's Responses to Questions

**Comments to the Author**

1. Is the manuscript technically sound, and do the data support the conclusions?

Reviewer #1: Yes

Reviewer #2: Yes

2. Has the statistical analysis been performed appropriately and rigorously? 

Reviewer #1: Yes

Reviewer #2: Yes

3. Have the authors made all data underlying the findings in their manuscript fully available?

Reviewer #1: Yes

Reviewer #2: Yes

4. Is the manuscript presented in an intelligible fashion and written in standard English?

Reviewer #1: Yes

Reviewer #2: Yes

5. Review Comments to the Author

Reviewer #1: Thank you for inviting me to review this comprehensive and interesting piece of work. Covid-19 has specifically devastated global well-being and all walks of life in 2020 and 2021. In this regard, this study has an essential and conducted critical study design. However, I want to add some sticky notes that will enhance the work's acceptability and quality and must be addressed before recommending it for publication.

1) A term used in abstract IMD, what is it? Be careful using abbreviations in the abstract and the main body of the paper before using their full form.

2) Please elaborate on the rationale behind using the ethnic code and their selection. Why did authors make a separate code for non-white British, Pakistani heritage, and others? Why didn't they consider the other ethnicity separately? Is it a focused study on Pakistani heritage mothers?

3) Literature on stringent lockdown measures must also be added and compared with the UK government because the survey was conducted during this phase. In this regard, a couple of studies suggested.

10.1016/j.heliyon.2021.e05912

10.5055/jem.0529

10.5055/jem.0621

10.1080/00223980.2022.2039891

4) The discussion section needs further work. Add relevant previous studies and compare their findings with your study results to show the study gaps and innovative points more clearly.

5) Implication of the study findings and conclusions also needs more convincing arguments related to their practical implications and highlighting the innovative points of the study for policymakers.

Reviewer #2: Dear authors,

Kindly find below few comments to be considered:

1- In the abstract, add clearly the aim of the study, methodology, and recommendations.

2- After the introduction, you have to add "Literature review".

3- Since your used Survey to emphasize the research hypotheses or questions. In your study i did not see clear research hypotheses or questions. Authors should add Literature review and at the end of the literature review you should generate research hypotheses or questions. Then in the results section you should show us how your results confirm or refute the hypotheses or questions of the study.

4- In the discussion section, you can laso show how the other studies support your results.

5- In the results section, you should also clarify statistically how you can enphasize the hypotheses or questions and which statistics support each one.

6- Usually the survey statistics should answer a research questions or hypotheses. I cannot see any of this relations.

All ther best,

6. PLOS authors have the option to publish the peer review history of their article (what does this mean?). If published, this will include your full peer review and any attached files.

Reviewer #1: No

Reviewer #2: No

---

## [Author Response · Author response to Decision Letter 0]

16 Jun 2023

We have carefully worked through the thoughtful comments of the editorial and reviewer teams and have provided our response to these comments for your consideration below. 

Thank you for the feedback and for signposting us to the relevant formatting guidance. We have now amended the manuscript and relevant files as per the PLOS ONE style templates. 

2. Please ensure that you have specified (1) whether consent was informed and (2) what type you obtained (for instance, written or verbal, and if verbal, how it was documented and witnessed). If your study included minors, state whether you obtained consent from parents or guardians. If the need for consent was waived by the ethics committee, please include this information.

We have specified this information under the ‘Ethics’ heading found within the ‘Methods’ section of the manuscript. 

We apologise for the disparity in the information provided in the ‘Funding Information’ and ‘Financial Disclosure’ sections. We have included a full financial disclosure in the manuscript which now correlates with the ‘Funding Information’ found within the editorial manager. 

We have provided a more detailed data availability statement in the editorial manager. Hopefully this is more helpful but please do let us know if you require further information. 

5. Please note that in order to use the direct billing option the corresponding author must be affiliated with the chosen institute. Please either amend your manuscript to change the affiliation or corresponding author, or email us at plosone@plos.org with a request to remove this option.

Thank you for this guidance. We have contacted plosone@plos.org to request that this option is now removed. 

We have specified this information in full under the ‘Ethics’ heading, found within the ‘Methods’ section of the manuscript. 

Reviewers' comments:

Reviewer's Responses to Questions

Comments to the Author

1. Is the manuscript technically sound, and do the data support the conclusions?

Reviewer #1: Yes

Reviewer #2: Yes

2. Has the statistical analysis been performed appropriately and rigorously?

Reviewer #1: Yes

Reviewer #2: Yes

3. Have the authors made all data underlying the findings in their manuscript fully available?

Reviewer #1: Yes

Reviewer #2: Yes

4. Is the manuscript presented in an intelligible fashion and written in standard English?

Reviewer #1: Yes

Reviewer #2: Yes

5. Review Comments to the Author

Reviewer #1: Thank you for inviting me to review this comprehensive and interesting piece of work. Covid-19 has specifically devastated global well-being and all walks of life in 2020 and 2021. In this regard, this study has an essential and conducted critical study design. However, I want to add some sticky notes that will enhance the work's acceptability and quality and must be addressed before recommending it for publication.

1) A term used in abstract IMD, what is it? Be careful using abbreviations in the abstract and the main body of the paper before using their full form.

Many thanks for identifying this error. We have corrected this, spelling out the abbreviated term in full. Furthermore, we have ensured the remainder of the abstract and main body of the paper has abbreviations spelled out in full before using their abbreviated terms. 

2) Please elaborate on the rationale behind using the ethnic code and their selection. Why did authors make a separate code for non-white British, Pakistani heritage, and others? Why didn't they consider the other ethnicity separately? Is it a focused study on Pakistani heritage mothers?

Thank you for this feedback. We have provided greater clarity behind the rationale for using these ethnic codes and why ethnicity was not considered separately in the manuscript.

3) Literature on stringent lockdown measures must also be added and compared with the UK government because the survey was conducted during this phase. In this regard, a couple of studies suggested.

10.1016/j.heliyon.2021.e05912

10.5055/jem.0529

10.5055/jem.0621

10.1080/00223980.2022.2039891

Thank you for your feedback and for the literature recommendations. Despite the Bradford population containing many mothers of Pakistani Heritage, we feel that the geographical and political setting of the research being limited to the UK government means that a comparative analysis of lockdown measures is perhaps not relevant to the findings of this paper at this time. However, we have included further references to literature exploring the scope and impact of the lockdown measures on this population in the UK. 

4) The discussion section needs further work. Add relevant previous studies and compare their findings with your study results to show the study gaps and innovative points more clearly.

Thank you for this feedback. We have hopefully addressed this point with further reflection upon the results and comparison to other study results, as recommended by both reviewers.

5) Implication of the study findings and conclusions also needs more convincing arguments related to their practical implications and highlighting the innovative points of the study for policymakers.

We have also addressed this in the manuscript based on the recommendations. 

Reviewer #2: Dear authors,

Kindly find below few comments to be considered:

1- In the abstract, add clearly the aim of the study, methodology, and recommendations.

Thank you for this feedback. We have amended the abstract to clearly add the study, methodology and recommendations. 

2- After the introduction, you have to add "Literature review".

We have more clearly defined the literature review, with the inclusion of the ‘Literature Review’ heading within the introduction. 

3- Since your used Survey to emphasize the research hypotheses or questions. In your study i did not see clear research hypotheses or questions. Authors should add Literature review and at the end of the literature review you should generate research hypotheses or questions. Then in the results section you should show us how your results confirm or refute the hypotheses or questions of the study.

Thank you for the feedback. We have reworked the introduction to improve the structure and signposting of the introduction. We have also reworked sections of the introduction to more clearly explain the aim and research questions of the study. 

4- In the discussion section, you can laso show how the other studies support your results.

Thank you for this feedback. We have hopefully addressed this point with further reflection upon the results and comparison to other study results, as recommended by both reviewers. 

5- In the results section, you should also clarify statistically how you can enphasize the hypotheses or questions and which statistics support each one.

Thank you for your feedback, we have amended the results section headings to make this clearer. 

6- Usually the survey statistics should answer a research questions or hypotheses. I cannot see any of this relations.

Thank you for your feedback. Hopefully the amendments we have made along the lines of the feedback above have addressed this point to provide greater clarity regarding the research questions and how the research answers these questions.

---

## [Decision Letter · Decision Letter 1]

30 Oct 2023

PONE-D-23-05276R1The long-term impact of the Covid-19 pandemic on financial insecurity in vulnerable families: findings from the Born in Bradford Covid-19 longitudinal study.PLOS ONE

Dear Dr. Reece,

Thank you for submitting your manuscript to PLOS ONE. After careful consideration, we feel that it has merit but does not fully meet PLOS ONE’s publication criteria as it currently stands. Therefore, we invite you to submit a revised version of the manuscript that addresses the points raised during the review process.

Minor revision

We look forward to receiving your revised manuscript.

Kind regards,

Wajiha Haq, PhD

Academic Editor

PLOS ONE

Journal Requirements:

Reviewers' comments:

Reviewer's Responses to Questions

**Comments to the Author**

1. If the authors have adequately addressed your comments raised in a previous round of review and you feel that this manuscript is now acceptable for publication, you may indicate that here to bypass the “Comments to the Author” section, enter your conflict of interest statement in the “Confidential to Editor” section, and submit your "Accept" recommendation.

2. Is the manuscript technically sound, and do the data support the conclusions?

Reviewer #1: Yes

3. Has the statistical analysis been performed appropriately and rigorously? 

Reviewer #1: Yes

4. Have the authors made all data underlying the findings in their manuscript fully available?

Reviewer #1: Yes

5. Is the manuscript presented in an intelligible fashion and written in standard English?

Reviewer #1: Yes

6. Review Comments to the Author

Reviewer #1: As I Recommended in the previous review, please add some stringency measures in other countries or regions and compare them with your findings, which have not been considered. The purpose behind suggesting such measures is to compare what others have done and how the UK government would have done it better or effectively. If they also do the same standard measures, what is the difference, and how would you say your findings are innovative or novel?

There is no need to summarize the key findings in the discussion section. It would be a repetition of the summary with a conclusion section. Just mention this summary while concluding remarks.

7. PLOS authors have the option to publish the peer review history of their article (what does this mean?). If published, this will include your full peer review and any attached files.

Reviewer #1: No

---

## [Author Response · Author response to Decision Letter 1]

1 Nov 2023

Thank you once again for your consideration of this manuscript. We have carefully worked through the thoughtful comments of the editorial and reviewer teams and have provided our response to these comments for your consideration in the attached 'Response to Reviewers' file. There were two points raised by Reviewer 1 and these have been addressed in the manuscript.

---

## [Editor Report · Decision Letter 2]

15 Nov 2023

The long-term impact of the Covid-19 pandemic on financial insecurity in vulnerable families: findings from the Born in Bradford Covid-19 longitudinal study.

PONE-D-23-05276R2

Dear Dr. Reece,

We’re pleased to inform you that your manuscript has been judged scientifically suitable for publication and will be formally accepted for publication once it meets all outstanding technical requirements.

Kind regards,

Wajiha Haq, PhD

Academic Editor

PLOS ONE
---

## [Editor Report · Acceptance letter]

17 Nov 2023

PONE-D-23-05276R2 

The long-term impact of the Covid-19 pandemic on financial insecurity in vulnerable families: findings from the Born in Bradford Covid-19 longitudinal study. 

Dear Dr. Reece:

I'm pleased to inform you that your manuscript has been deemed suitable for publication in PLOS ONE. Congratulations! Your manuscript is now with our production department. 

Kind regards, 

on behalf of

Dr. Wajiha Haq 

Academic Editor

PLOS ONE